behaviour/ecology

collective personality, repeatability, plasticity, reaction norm

**Author for correspondence:**
Hannah E. A. MacGregor
e-mail: hannah.macgregor@bristol.ac.uk

# Collective motion diminishes, but variation between groups emerges, through time in fish shoals

## Hannah E. A. MacGregor and Christos C. Ioannou

School of Biological Sciences, University of Bristol, Bristol BS8 1TQ, UK

HEAM, 0000-0001-5379-8392; CCI, 0000-0002-9739-889X

Despite extensive interest in the dynamic interactions between individuals that drive collective motion in animal groups, the dynamics of collective motion over longer time frames are understudied. Using three-spined sticklebacks, *Gasterosteus aculeatus*, randomly assigned to 12 shoals of eight fish, we tested how six key traits of collective motion changed over shorter (within trials) and longer (between days) timescales under controlled laboratory conditions. Over both timescales, groups became less social with reduced cohesion, polarization, group speed and information transfer. There was consistent inter-group variation (i.e. collective personality variation) for all collective motion parameters, but groups also differed in how their collective motion changed over days in their cohesion, polarization, group speed and information transfer. This magnified differences between groups, suggesting that over time the 'typical' collective motion cannot be easily characterized. Future studies are needed to understand whether such between-group differences in changes over time are adaptive and represent improvements in group performance or are suboptimal but represent a compromise between individuals in their preferences for the characteristics of collective behaviour.

## 1. Introduction

Collective motion is widespread in living systems from bacteria to humans [1,2]. In these systems, there is no centralized control of group movement, yet highly synchronized behaviours can arise from local interactions between individuals, even when individuals lack an awareness of the position and movement of all group members [3]. Parallels are often drawn between animal collective behaviour and physical and chemical systems during phase transitions [4,5]; however, the traits of animal

groups (e.g. their speed, direction and spatial organization) differ fundamentally from non-living systems in that they emerge from interactions between complex and biologically motivated individuals and are shaped by natural selection [6]. Although there are costs to animal collective behaviour as it can result in competition for food [7] and exploitation by predators and parasites [8,9], the independent evolution and diversity of collective behaviours across animal taxa suggest that benefits from improved predator avoidance [10] and use of resources [11] often outweigh the costs.

Shifts in the benefits and costs of collective behaviour can result in changes over evolutionary time [12–14]. Over within-generation timescales, collective behaviour can change in response to changing environmental conditions, such as cues from predators or food, or abiotic factors such as habitat structure [15–20]. Our current understanding of collective behaviour is, however, incomplete from a developmental perspective because there is limited information on the behavioural changes that occur in animal collectives over short timescales without modification to environmental conditions [21–25]. The increased attraction between individuals during early development is one such mechanism by which this can occur [23,26]. Alternatively, changes in collective behaviour can be associated with repeated exposure to the same environmental conditions, such as acclimatization to an environment that was initially novel. For example, zebrafish shoals spend an increasing proportion of time in a disordered compared to an ordered state as they habituate over time to a novel environment [27].

Different groups within the same population can also consistently differ in their collective behaviour over time and contexts. This group-level variation is akin to the consistent behavioural variation between individuals described as animal personality variation [28]. Evidence for consistent differences between groups, often referred to as group or collective personality variation, is strongest in social insects in behaviours associated with group defence and foraging (e.g. [29,30]) and has been linked to colony productivity and survival [31]. There are also examples in vertebrates, for instance, in both chimpanzees [32] and fish [33].

Not only do individuals vary in their average levels of behaviour, but they may also vary in their change in behaviour over time or contexts; i.e. they vary in their behavioural plasticity. Individual red knots, for example, differ in their change in vigilance behaviour in response to predation risk [34]. Differences in plasticity can be quantified using a behavioural reaction norm approach where random slope parameters in regression models are free to vary among individuals or groups [35–37]. While reaction norm approaches have been applied widely to study variation in plasticity between individuals, they have not been used to examine differences between groups in the plasticity of collective behaviour. Examination of random slope variation is, however, key to understanding the emergence and maintenance of consistent group-level differences in behaviour, because differences between groups may be lost or magnified over time.

It thus remains unclear when variation between groups is apparent; inter-group differences could establish early in group formation, and be stable or diminish over time, or they could develop gradually. In three-spined sticklebacks, fish form shoals of unrelated individuals [38] and group membership may remain relatively stable over time [39]. Here, we tested 12 groups of eight sticklebacks repeatedly in the same context (swimming in an open arena) under controlled laboratory conditions. The groups were tested over multiple days up to 12 times over four weeks. Each trial lasted approximately 25 min and group behaviour was analysed during five 2.5 min time intervals that were separated by the presentation of a food item (see Material and methods, and [7] for analyses of responses to the food item presentations). We used high-resolution video tracking to examine changes over time in six traits of collective motion that are known to be functionally important: polarization, which measures the alignment of individuals between 0 (no alignment) and 1 (complete alignment, where higher levels of polarization can improve social information transfer and reduce the risk of predation due to confusion effects: [40,41]), the speed of the group centroid (where group speed can affect responses to predators and encounter rates with food: [14]), convex hull area as a measure of spacing between individuals (where less cohesive groups are at greater risk of predation: [42]), the rate of transitions in state between polarized (aligned) and unpolarized (swarm-like) states [43,44], the maximum cross-correlation in speed (which is a measure of information transfer: [45]), and the rate of switches in leadership over time (where individuals at the front of the group have greater access to foraging rewards at a cost of greater risk of predation, and increased switching is observed in shoals from high-predation habitats: [10,46–48]). Using a reaction norm approach combined with model selection procedures [35,49], we then determined whether the groups differed from one another in the dynamics of their collective behaviour over shorter (within trials) and longer (between days) timescales.

# 2. Material and methods

## 2.1. Study animals

Three-spined sticklebacks (*Gasterosteus aculeatus*, $27 \pm 2.4$ mm, mean $\pm$ s.d., standard body length at the time of testing for $n = 96$ individuals) were collected from the River Cary, Somerset, UK (grid ref: ST 469 303) in September 2016 and transported to fish facilities at the University of Bristol, UK. The fish were housed in three glass tanks (70 cm (L) $\times$ 45 cm (W) $\times$ 37.5 cm (H)) of approximately 50 individuals for 10 months before testing and were fed daily with brine shrimp or defrosted frozen bloodworm (*Chironomid* sp. larvae). The photoperiod was a 11 : 13 h light : dark cycle and ambient temperature was maintained at 16°C to prevent the fish from entering reproductive condition (fish were not sexed). One week prior to the experiment, we assigned the fish to 12 groups of eight individuals in a randomized complete block design by repeatedly netting 12 individuals from a tank and randomly allocating each individual to one of 12 groups. This minimized any potential differences between groups in case the netting of individuals from the holding tanks was biased towards sampling individuals with particular traits [50]. The fish were given 6 days to habituate in their groups in smaller glass holding tanks (70 (L) $\times$ 25 (W) $\times$ 37.5 (H) cm) enriched with a horizontal piece of PVC tubing and an artificial plant prior to the first trial.

## 2.2. Experimental procedure

The groups were tested repeatedly up to 12 times over four consecutive weeks. Each group was tested every other day from Monday to Friday. Six groups were tested per day, therefore at the start of each week, the 12 groups were randomly allocated to two sets of six groups and the set that would have trials on Monday, Wednesday and Friday was determined at random. Trials took place between 09.45 and 16.00 each day and the order of testing of the groups within a day was randomized. Each trial was conducted in a white oval-shaped open experimental arena (133.5 (L) $\times$ 72 (W) $\times$ 62 (wall height) cm) with a water depth of 10 cm which was maintained at the same temperature as the holding tanks. Because individuals may seek refuge in heterogeneous environments, no artificial enrichment was provided in the arena to encourage collective movement. There was a small amount of variation in the length of each trial ($25 \pm 2.2$, mean $\pm$ s.d. trial length in minutes for 84 trials) due to the experimental protocol (see [7] for further details). Trials were filmed from above with a Panasonic HC-VX980 video camera in 4K (3840 $\times$ 2178 pixels) and a temporal resolution of 25 frames per second. After each trial, the fish were returned to their holding tank and were fed with bloodworm following the final trial of each day. Trials were terminated for a group before the end of the fourth week if any individual in the group began displaying signs of poor health.

At the start of a trial, all eight fish were netted into the centre of the arena and allowed 2 min to acclimatize. For the part of the experiment analysed and reported in [7], a red-tipped pipette that delivered a single food item (a defrosted bloodworm) was presented to each group on six occasions during each trial. The time interval between presentations was approximately 4 min ($4.3 \pm 1.0$ min, mean $\pm$ s.d.), which allowed the group to resume normal swimming behaviour. In the present study, we analysed the video footage of the groups' behaviour in the 2.5 min immediately prior to the second to sixth presentations of the red-tipped pipette in each trial to obtain trajectory data for five consecutive time intervals (first–fifth) per trial (i.e. 12.5 min of video footage per trial). Data for three intervals from two separate trials were not obtained due to corrupted video files. During the first week of trials, one individual in each of three groups was replaced (one due to injury and two deaths of unknown cause) with an individual that was naive to the experiment and given 24 h to habituate within their group in the holding tank prior to testing. All trials conducted prior to these replacements were included in the analyses.

## 2.3. Data processing

Video files were converted to MPEG-4 HD (1920 $\times$ 1080 pixels) in Handbrake (v. 1.0.7, https://handbrake.fr/) where 1 mm was equal to 2.7 pixels. Trajectory data for each fish were obtained using idTracker v. 2.1 [51]. Only frames with complete trajectory information for all eight individuals in the groups were included in analyses. Frames where the speed of any individual in the group exceeded approximately 46 cm s$^{-1}$ (50 pixels per frame) were excluded from further analyses because visual

inspection of the data revealed that these were probably due to tracking errors. The remaining trajectory data were then smoothed using a Savitzky–Golay filter with a span of approximately 0.5 s (13 frames) and a polynomial of 3 degrees in R package *Trajr* [52]. One interval in a trial had less than 30 s of trajectory information after data processing due to poor tracking quality and was excluded. The final dataset consisted of trajectory information for 416 intervals (3705 ± 173, mean ± s.d. frames per interval) from 84 trials across the 12 groups.

For analyses, we calculated the median (across frames for each interval per trial) of the groups' polarization, centroid speed (based on the distance travelled since the previous frame) and the convex hull area due to the right skew in the distributions of these variables. For each interval, we also calculated the maximum cross-correlation in the speed of an individual to their nearest neighbour (with a maximum lag of 300 frames (12 s)) and used the mean maximum cross-correlation in speed across the eight individuals for analyses. The rate of transitions in state was quantified by the number of transitions of the group between polarized (greater than 0.65) and unpolarized (less than 0.65) states divided by the total number of frames in the time interval (excluding time steps with missing trajectory data). Finally, we quantified the rate of switches in leadership as the number of switches of each fish to or from the lead position in the group (determined by an individual's rank distance along the direction of group motion (first–eighth)) between consecutive frames when the group was in a polarized (greater than 0.65) state divided by the number of frames. We used the mean rate of switches in leadership across the eight individuals per interval per trial for analyses (see electronic supplementary material for further details on the quantification of the six collective traits).

## 2.4. Data analyses

All data analyses were carried out in R v. 3.6.2 [53]. To test for an overall effect of time on collective motion, each model included interval (first–fifth) and day (1st–12th) as continuous main effects and group identity as a random intercept. Convex hull area was square-root transformed to meet parametric model assumptions. Gaussian models were fitted using restricted maximum likelihood, and the significance of interval and day were tested using Kenward–Roger approximation F-tests in R package *lmerTest* [54]. The significance of the effects of interval and day on the rate of switches in leadership was tested with likelihood ratio tests with models fitted using maximum likelihood. Residuals from all models were checked visually for conformity to assumptions of homogeneity of variance and normality of error.

To determine whether the groups differed on average in their collective motion, the performance of models fitted with and without group identity as a random intercept (which accounts for average group-level variation) was compared using the conditional AIC (cAIC) [55] and corrected AIC (AICc), respectively. When the model with group identity was greater than two Δ(c)AIC(c) units from the model without, this was deemed as support for inter-group differences in behaviour [49]. cAIC controls for the number of effective model parameters accounting for shrinkage in the random effects [55] and was estimated in R package *cAIC4* [56]. For models without group identity (i.e. no random effects), the AICc of the model estimated in R package *AICcmodavg* [57] was used for comparison. For switches in leadership, models were compared based on AICc because cAIC estimates for beta regression models were unavailable.

Random slopes are interactions between fixed effect covariates and random effects and represent reactions of each groups' collective behaviour over time [58]. To test whether the groups differed in their trajectories of change over time, the performance of models fitted with and without interval and day as random slopes (which account for group-level behavioural plasticity within trials and between days, respectively) were compared using the differences in cAIC. Three models (table 1) were not included in model comparisons because (c)AIC(c) estimates were unreliable due to model overfitting that was indicated by estimated perfect correlations between random effect terms [59]. All models were fitted using maximum likelihood. Random slope variation disrupts repeatability which will vary depending on the level of the covariate; therefore, we do not report inter-group repeatability estimates.

# 3. Results

To test for an effect of time on collective motion, each movement trait was analysed as a response variable in a separate linear mixed model, except for the rate of switches in leadership, which was analysed in a generalized linear mixed model fitted with a beta distribution. As time progressed both within trials (increases in interval) and between days, the groups became slower, less polarized and less cohesive

**Table 1.** Conditional AIC (cAIC) and corrected AIC (AICc) scores for models with different random effect structures (with and without group identity as a random intercept and with and without interval and day as random slopes). All models included interval and day as main effects. Estimated degrees of freedom (d.f.), Δ(c)AIC(c) from the (c)AIC(c) of the best-supported model (in italics) and model rank (where 1 is the best-supported and 4 the least-supported model) are reported for each trait. AIC is estimated for models with no random effects and used in the comparison of models of the rate of switches in leadership because cAIC is unavailable for beta regressions. Convex hull area was square-root transformed to meet parametric assumptions. Missing values are where models were unable–slope correlation parameters and therefore produced unreliable (c)AIC(c) scores.

| trait | random intercept | random slope | (c)AIC(c) | d.f. | Δ(c)AIC(c) | rank |
|---|---|---|---|---|---|---|
| polarization | none | none | −598.29 | 4 | 267.25 | 4 |
| | group identity | none | −731.37 | 14.4 | 134.17 | 2 |
| | group identity | interval | −726.29 | 17.6 | 139.25 | 3 |
| | *group identity* | *day* | *−865.54* | *24.0* | *0* | *1* |
| centroid speed (mm s⁻¹) | none | none | 3290.02 | 4 | 321.21 | 4 |
| | group identity | none | 3123.53 | 14.6 | 154.72 | 2 |
| | group identity | interval | 3126.19 | 19.7 | 157.38 | 3 |
| | *group identity* | *day* | *2968.80* | *24.5* | *0* | *1* |
| convex hull area (mm) | none | none | 4249.39 | 4 | 323.27 | 3 |
| | group identity | none | 4200.89 | 14.5 | 174.86 | 2 |
| | group identity | interval | — | — | — | — |
| | *group identity* | *day* | *3926.03* | *24.7* | *0* | *1* |
| rate of transitions in state (frames⁻¹) | none | none | −2848.13 | 4 | 124.70 | 3 |
| | group identity | none | −2968.47 | 14.3 | 4.28 | 2 |
| | *group identity* | *interval* | *−2972.75* | *20.6* | *0* | *1* |
| | group identity | day | −2968.47 | 19.3 | 4.28 | 2 |
| maximum cross-correlation in speed | none | none | −1056.45 | 4 | 226.88 | 4 |
| | group identity | none | −1177.27 | 14.3 | 106.16 | 2 |
| | group identity | interval | −1171.56 | 17.5 | 111.87 | 3 |
| | *group identity* | *day* | *−1283.43* | *23.6* | *0* | *1* |
| rate of switches in leadership (frames⁻¹) | none | none | −5085.29 | 4 | 16.72 | 2 |
| | *group identity* | *none* | *−5102.01* | *5* | *0* | *1* |
| | group identity | interval | — | — | — | — |
| | group identity | day | — | — | — | — |

on average. Individuals also became less responsive to their near neighbours, with lower maximum cross-correlations in speed (electronic supplementary material, table S1). Although the groups transitioned between polarized and unpolarized states significantly more at the end compared to the start of the trials, this effect was not observed between days (electronic supplementary material, table S1). There was no evidence for an overall change in switches in leadership over time either within trials or between days (electronic supplementary material, table S1).

To determine whether there were consistent differences in the average value of each trait between groups, we compared the performance of models fitted with and without group identity as a random intercept (table 1). Based on a difference in Akaike information criterion of greater than two units [49,55], models including group identity were more likely given the data than models without for all traits, suggesting that the groups differed in their collective motion (table 1).

There were also differences between the groups in how their collective motion changed over time (figure 1a–e). Comparing the performance of models with and without day and interval as random slope parameters, the most likely models explaining variation in polarization, centroid speed, convex hull area and maximum cross-correlation in speed included day as a random slope (table 1 and figure 1a–d). While most groups showed less social collective behaviour for these traits as the days progressed (decreased polarization, group speed and speed correlation, with an increase in convex hull area), some showed no change (figure 1a–d). As a result, the model-estimated variance between the groups also increased over the days of the experiment, i.e. the differences between groups became more pronounced (figure 1f–i). Groups did not differ in how they changed over time within trials in these traits (table 1). By contrast, the best-supported model for the rate of transitions in the state included interval as a random slope, showing that the groups differed in their response over a short timescale (figure 1e) and the model-estimated variance between the groups also increased over time within trials (figure 1j).

## 4. Discussion

Differences between groups in their collective behaviour may arise due to differences between individuals that are present when groups first form and then remain stable over time. Alternatively, variation between groups may become stronger as individuals in newly formed groups have opportunities to gather information and socially interact [21]. Our data support the latter scenario with more variation between the groups in collective motion at the end compared to the start of the experiment. The groups displayed different rates of change in their collective motion, i.e. differing degrees of plasticity, with variation in the strength, and in some cases direction, of their response as the days of the experiment progressed. Previous work has reported consistent group differences across contexts in alignment, speed and cohesiveness in three-spined sticklebacks ([33] although see [61]). We show that differences in these traits, as well as the maximum cross-correlation in speed, between shoals are magnified over several weeks when group membership is stable. Positive feedback through repeated interactions between the same individuals and learning can exaggerate initially small differences over time [21,22]. Social conformity, where groups converge in their behaviour towards the mean behaviours of all individuals [62] or towards that of the most influential individuals [25,63], is also predicted to increase between-group variation. This may explain how differences between groups in their collective behaviour over time may emerge from minimal between-group heterogeneity, which we ensured with the method used to assemble the groups. Since we do not know the sex ratio composition of the groups, it is also possible that sex-specific interaction patterns explain some of the divergence in behaviour [64].

Despite these differences between groups in their responses over time, overall, the groups tended to become slower, less cohesive and less polarized with reduced cross-correlation in speed both within trials and between days of testing. This is consistent with a weakening of attraction and responsiveness to near neighbours. Functionally, these temporal changes are associated with a reduction in social information sensing and anti-predator benefits [7,41,65] and suggest that the groups were acclimatizing over time to the low-risk, predator-free context. Maintaining social responsiveness during collective motion by forming fast, cohesive and highly polarized groups may be cognitively demanding and energetically costly [66]. Therefore, in the absence of environmental pressures to preserve these traits of shoaling, animals become less collective in their movement [27] and decision making [25]. Previous studies have also highlighted a role for increased aggression and familiarity. In naturally clonal Amazonian mollies, for example, inter-individual distances were found to be higher in groups of more familiar individuals, which was associated with increased aggression [67]. In species that form social

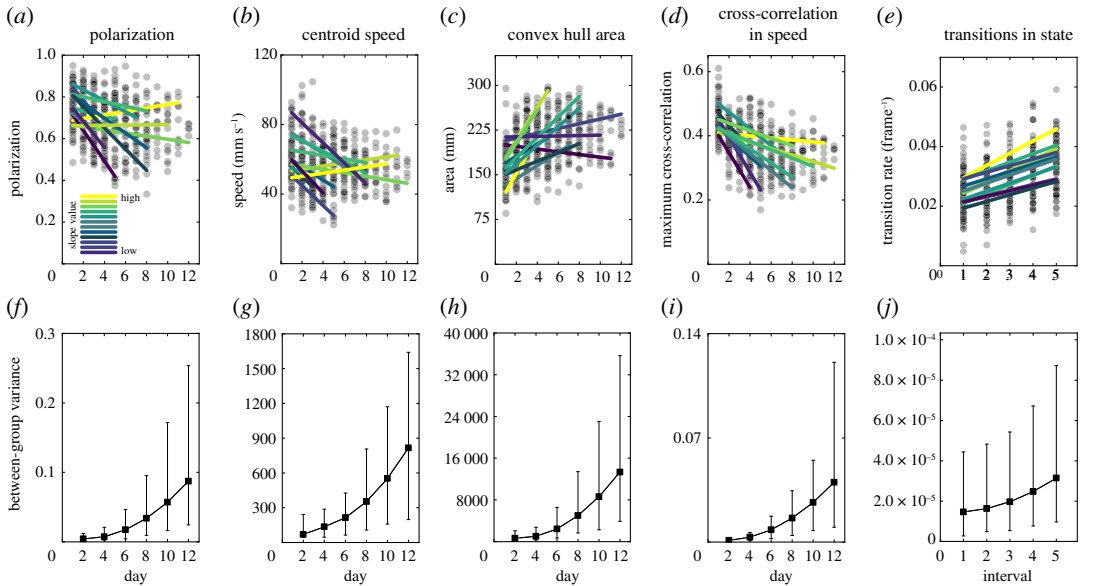

**Figure 1.** Variation between the groups in collective motion over time; (*a,d*) show the predicted reaction norms of the 12 groups across days from the best-supported models to explain polarization, centroid speed, convex hull area and maximum cross-correlation in speed, and (*e*) shows the predicted reaction norms from intervals during a trial from the best-supported model to explain the rate of transitions in state. The predicted reaction norms for each group are coloured by the slope (highest (yellow) to lowest (dark purple)) within each panel and estimated using the *predict* function in R package *lme4* [60]. For predictions, the interval (in *a–d*) and day (in *e*) are held at their mean value. The circles are the raw values (i.e. per interval per trial, $n = 416$). For the convex hull area, values are square-root transformed to meet parametric assumptions. In (*a–d*), the lines are truncated to the day where the experiment ended for a particular group; (*f–i*) show the trends in between-group variance over days of the experiment. Between-group variance is the model-estimated variance in the random intercept of group identity when the variable is centred at day two, four, six, eight, 10 and 12, respectively, shown as filled squares with upper and lower 95% bootstrap confidence intervals shown as vertical lines for each estimate; (*j*) shows the trend in between-group variance in the rate of transitions in state within trials for the model-estimated variance in the random intercept of group identity when the variable interval is centred at interval one to five, respectively. The trend in the model-estimated variance in group identity reveals how inter-group differences in collective motion change over time.

hierarchies, a reduction in collective behaviour over time could also be driven by the establishment of social dominance [64].

Studies of animal collective motion typically only provide a snapshot of group behaviour. Demonstrating that groups show variation in their plasticity over time has significant implications for the study of collective behaviour as this magnifies variation between groups, making the detection of differences between treatments [61], populations (for example along with an environmental gradient; [47]) or species [68] more difficult. A greater number of independent experimental groups may be required than originally thought, and reliance on a few replicate groups may be inadequate. Testing newly assembled groups may alleviate this, although this may not maximize ecological relevance if the animals have relatively stable social memberships. The time available for behaviour between groups to diverge will be determined by the degree to which social membership is stable, which in turn may be driven by ecological factors such as population density (determining the rate at which groups encounter one another and can exchange group members: [69]) and predation risk (determining the likelihood individuals change groups: [70,71]). A more holistic understanding of collective motion would consider such ecological factors, as well as the development of collective behaviour over time.

Ethics. All procedures regarding the use of animals followed United Kingdom guidelines and were approved by the University of Bristol ethics committee (UIN UB/17/060).

Data accessibility. The dataset and code accompanying this paper can be downloaded from Dryad Digital Repository: https://doi.org/10.5061/dryad.547d7wm8p [72].

Authors' contributions. C.C.I. and H.E.A.M. conceived and designed the study. H.E.A.M. conducted the experiment and collected and analysed the data. H.E.A.M. and C.C.I. interpreted the data. H.E.A.M. wrote the first draft of the manuscript and both authors wrote and approved the final version.

Competing interests. We declare we have no competing interests.

Funding. This research was funded by a Natural Environment Research Council grant no. NE/P012639/1 awarded to C.C.I. H.E.A.M. gratefully acknowledges support from The Association for the Study of Animal Behaviour.

Acknowledgements. We thank David Bierbach and one anonymous reviewer for their valuable comments which helped to improve the manuscript.

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
