## [Peer Review File · Royal Society Open Science]

Review History

RSOS-210655.R0 (Original submission)

Review form: Reviewer 1

Is the manuscript scientifically sound in its present form?

Yes

Are the interpretations and conclusions justified by the results?

Yes

Is the language acceptable?

Yes

Do you have any ethical concerns with this paper?

No

Have you any concerns about statistical analyses in this paper?

No

Recommendation?

Accept with minor revision (please list in comments)

Comments to the Author(s)

The manuscript „Emergence through time of variation between groups in fish shoal collective motion“ describes how collective measures of shoaling in groups of sticklebacks change over time and how groups diverge in this regard. The paper makes the point that newly assembled groups may be more similar towards each other and over time, observable behavior will diverge greatly. I really enjoyed reading the paper and I did not find any major points of critique. Well done! I was only wondering why the authors do not include a discussion on increasing familiarity among group members as an explicit explanation. For example, my own research on Poeciliid fishes showed that social hierarchies emerge when individuals are kept together for longer times and these are often sex-specific (see (Bierbach, Oster et al. 2014)). Furthermore, we found a more or less universal decrease in cohesion among shoal members over time (although we did not look at among-group variation on this). This effect was also found in all-female groups where we could further show that increasing levels of aggressions may cause to decrease in cohesion (Doran, Bierbach et al. 2019). While I do not expect the authors to explicitly cite these papers, a more in-depth discussion on the causes of the decreased cohesion and polarization (possible cause: aggressiveness and hierarchies) as well as group differences (possible cause: sex-specific interaction patters even outside the breeding season) may be helpful for the reader as the similar observations have been made before.

David Bierbach

Bierbach, D., S. Oster, J. Jourdan, L. Arias-Rodriguez, J. Krause, A. M. Wilson and M. Plath (2014). "Social network analysis resolves temporal dynamics of male dominance relationships."

Behavioral Ecology and Sociobiology: 1-11.

Doran, C., D. Bierbach and K. L. Laskowski (2019). "Familiarity increases aggressiveness among clonal fish." Animal Behaviour 148: 153-159.

Review form: Reviewer 2

Is the manuscript scientifically sound in its present form?

Yes

Are the interpretations and conclusions justified by the results?

Yes

Is the language acceptable?

Yes

Do you have any ethical concerns with this paper?

No

Have you any concerns about statistical analyses in this paper?

No

Recommendation?

Accept with minor revision (please list in comments)

Comments to the Author(s)

Dear Drs MacGregor and Ioannou,

I really enjoyed your paper; please see my detailed comments in the attached pdf (Appendix A).

Decision letter (RSOS-210655.R0)

Dear Dr MacGregor

On behalf of the Editors, we are pleased to inform you that your Manuscript RSOS-210655 "Emergence through time of variation between groups in fish shoal collective motion" has been accepted for publication in Royal Society Open Science subject to minor revision in accordance with the referees' reports. Please find the referees' comments along with any feedback from the Editors below my signature.

Please submit your revised manuscript and required files (see below) no later than 7 days from today's (ie 16-Aug-2021) date. Note: the ScholarOne system will 'lock' if submission of the revision is attempted 7 or more days after the deadline. If you do not think you will be able to meet this deadline please contact the editorial office immediately.

on behalf of Professor Kevin Padian (Subject Editor)
openscience@royalsociety.org

Reviewer comments to Author:

Reviewer: 1

Comments to the Author(s)

The manuscript „Emergence through time of variation between groups in fish shoal collective motion“ describes how collective measures of shoaling in groups of sticklebacks change over time and how groups diverge in this regard. The paper makes the point that newly assembled groups

may be more similar towards each other and over time, observable behavior will diverge greatly. I really enjoyed reading the paper and I did not find any major points of critique. Well done! I was only wondering why the authors do not include a discussion on increasing familiarity among group members as an explicit explanation. For example, my own research on Poeciliid fishes showed that social hierarchies emerge when individuals are kept together for longer times and these are often sex-specific (see (Bierbach, Oster et al. 2014)). Furthermore, we found a more or less universal decrease in cohesion among shoal members over time (although we did not look at among-group variation on this). This effect was also found in all-female groups where we could further show that increasing levels of aggressions may cause to decrease in cohesion (Doran, Bierbach et al. 2019). While I do not expect the authors to explicitly cite these papers, a more in-depth discussion on the causes of the decreased cohesion and polarization (possible cause: aggressiveness and hierarchies) as well as group differences (possible cause: sex-specific interaction patters even outside the breeding season) may be helpful for the reader as the similar observations have been made before.

David Bierbach

Bierbach, D., S. Oster, J. Jourdan, L. Arias-Rodriguez, J. Krause, A. M. Wilson and M. Plath (2014). "Social network analysis resolves temporal dynamics of male dominance relationships." *Behavioral Ecology and Sociobiology*: 1-11.
 Doran, C., D. Bierbach and K. L. Laskowski (2019). "Familiarity increases aggressiveness among clonal fish." *Animal Behaviour* 148: 153-159.

Reviewer: 2

Comments to the Author(s)

Dear Drs MacGregor and Ioannou,

I really enjoyed your paper; please see my detailed comments in the attached pdf ("RSOS_Review.pdf").

===PREPARING YOUR MANUSCRIPT===

===PREPARING YOUR REVISION IN SCHOLARONE===

Author's Response to Decision Letter for (RSOS-210655.R0)

See Appendix B.

Decision letter (RSOS-210655.R1)

Dear Dr MacGregor,

I am pleased to inform you that your manuscript entitled "Collective motion diminishes, but variation between groups emerges, through time in fish shoals" is now accepted for publication in Royal Society Open Science.

on behalf of Prof Kevin Padian (Subject Editor)
openscience@royalsociety.org

Appendix A

Review notes for “Emergence through time of variation between groups in fish shoal collective motion”

In this study, the authors examine how standard, important, reasonable quantitative measures of collective motion vary in time for shoals of three-spined sticklebacks. The study spans a large set of repeated observations of twelve groups of initially newly subgrouped shoals of eight fish, with observations of 12.5 minutes duration in total (drawn from a larger set of observations of approximately 25 minutes that included a regular feeding regime) of each group made up to twelve times over a period of four weeks. The nature of the data set means that variation in collective motion measures could be examined over relatively short time scales (from subsets of the 12.5 minutes of data recorded per experimental trial) and longer durations (from the repeated observations of groups over a four week period). The authors determined that over both the short and long time scales, cohesion (quantified by the area of the convex hull that contained all group members), group order (quantified by polarisation, a measure of overall group alignment), the speed of the group centroid and a proxy for information transfer (the maximum cross-correlation in speed) all diminished as time from the start of observations increased. I think these are extremely interesting results! In addition, the authors identified variation across groups in both measures of collective motion, and how these measures varied over time, which may have (or likely has) a connection to how the behaviours of individual group members influence and determine overall collective movements.

I really enjoyed working through this paper. I think the paper is well written, the analysis is well done, the results are interesting, and the results point to multiple avenues for further potential research. As such, I happily recommend that the manuscript be accepted for publication subject to addressing the following minor considerations.

Main text

Lines 1-2. I'm not sure that the title clearly captures the content of the paper as well as it could. Maybe something like “Group cohesion, order and information transfer diminish over time in fish shoals”, or more specifically “Group cohesion, order and information transfer diminish over time in shoals of three-spined sticklebacks (*Gasterosteus aculeatus*)” could be considered as alternatives?

Line 112. It's noted that fish were not sexed. Has there been any work to examine potential differences in the movement behaviour of male and female sticklebacks? Might the observed differences across groups be related to male/female composition?

Line 113. In relation to the group sizes, has any work be done on three-spined sticklebacks that examines the retention of individual movement traits or tendency to conform as a group size varies? I know that idiosyncrasies of the movement patterns of individuals in another species, eastern mosquitofish, become harder to identify as group size increases (see for example, Herbert-Read et al. (2013), Proc R Soc B 280(1752): 20122564) – could the variation across groups be explained by the groups being of a size that group members are not (yet) conforming to each other? Perhaps heterogeneity of group members in general movement behaviour (including the underlying dynamic rules that govern collective motion) could explain the variation across groups that has been observed? Theoretical support for this

idea comes from Romey (1996), *Ecological Modelling* 92:65-77, and empirical support from Jolles et al. (2017), *Current Biology* 27:2862-2868.

Lines 175-181 and bottom of Table 1 (on page 13 of the review version of the manuscript). Given that the results relating to switches in leadership taking into account time interval and day were not reliable, might it be reasonable to examine durations spent in the leadership position (the front of the group, which is a completely reasonable hypothesis) via survival analysis (for example, Kaplan-Meier survival curves with appropriate confidence intervals for visualisation, and log-rank tests for significance)?

Lines 314-315. I think there is a missing right parenthesis.

Supplementary materials and methods

I think the presentation of some of the equations could be improved.

At the second line under *Quantification of traits*, I suggest writing the individual coordinates as $(x_i(t), y_i(t))$ with t measured in time, rather than time steps.

(i) I suggest writing the equation for the polarisation as:

$$P(t) = \frac{1}{N} \left| \sum_{i=1}^N \vec{u}_i(t) \right|,$$

so that the fact that the direction vectors are a function of time is unambiguous.

(ii) Given that the authors have established notation for the duration between time steps/tracked video frames as Δt , I suggest writing the equation for the centroid speed as:

$$v_c(t) = \frac{\sqrt{(x_c(t) - x_c(t - \Delta t))^2 + (y_c(t) - y_c(t - \Delta t))^2}}{\Delta t}.$$

I suggest rewriting the equation for the group centroid slightly as:

$$\begin{pmatrix} x_c(t) \\ y_c(t) \end{pmatrix} = \frac{1}{N} \begin{pmatrix} \sum_{i=1}^N x_i(t) \\ \sum_{i=1}^N y_i(t) \end{pmatrix}.$$

(vi) I'm not sure about the formulation used by the authors here, although it's possible that it could just be due to some typos. The method for determining the front-to-back position of group members first involves for each individual, indexed i , determining the component of the vector from the group centroid to the position of individual i in the direction of the group velocity vector (as summarised by the equations for $\cos \theta$ and d). Based on the equation for the group speed given at (ii) in the supplementary methods, the authors (implicitly) used a backwards difference approximation for the x - and y -components of the group velocity vector (which is valid). Where I think there is an error is that the position references for both the group centroid and the position of individual i should be taken from the same time, t , but the equation for the dot product ($\cos \theta$) suggests that the authors constructed the vector from the group centre at step $t - 1$ (or time $t - \Delta t$) to the position of individual i at the next time step

(or just time) t . If this is what the authors did, then I suggest that these calculations might have to be redone/checked using the equations I suggest next. If this is just a typo, then I think that the equation for $\cos \theta$ could be rewritten as

$$\cos \theta = \frac{(x_i(t) - x_c(t))(x_c(t) - x_c(t - \Delta t)) + (y_i(t) - y_c(t))(y_c(t) - y_c(t - \Delta t))}{\sqrt{(x_i(t) - x_c(t))^2 + (y_i(t) - y_c(t))^2} \sqrt{(x_c(t) - x_c(t - \Delta t))^2 + (y_c(t) - y_c(t - \Delta t))^2}},$$

if my suggestion above about writing t as time rather than time step is applied. The rewriting of the coordinates of the fish is not needed again, if defined as I suggest at the beginning of the *Quantification of traits* section, and the equation for d could be written as

$$d = \cos \theta \sqrt{(x_i(t) - x_c(t))^2 + (y_i(t) - y_c(t))^2}.$$

The equation for the mean nearest neighbour distance on page 3 does not seem correct. Perhaps it could be written as:

$$NND = \frac{1}{N} \sum_{i=1}^N \min \left\{ \sqrt{(x_i(t) - x_j(t))^2 + (y_i(t) - y_j(t))^2} : j = 1, \dots, N, j \neq i \right\},$$

where $\min\{\}$ denotes the minimum of the set of values described within the braces?

Appendix B

Reviewer 1

The manuscript „Emergence through time of variation between groups in fish shoal collective motion“ describes how collective measures of shoaling in groups of sticklebacks change over time and how groups diverge in this regard. The paper makes the point that newly assembled groups may be more similar towards each other and over time, observable behavior will diverge greatly. I really enjoyed reading the paper and I did not find any major points of critique. Well done!

I was only wondering why the authors do not include a discussion on increasing familiarity among group members as an explicit explanation. For example, my own research on Poeciliid fishes showed that social hierarchies emerge when individuals are kept together for longer times and these are often sex-specific (see (Bierbach, Oster et al. 2014)). Furthermore, we found a more or less universal decrease in cohesion among shoal members over time (although we did not look at among-group variation on this). This effect was also found in all-female groups where we could further show that increasing levels of aggressions may cause to decrease in cohesion (Doran, Bierbach et al. 2019). While I do not expect the authors to explicitly cite these papers, a more in-depth discussion on the causes of the decreased cohesion and polarization (possible cause: aggressiveness and hierarchies) as well as group differences (possible cause: sex-specific interaction patters even outside the breeding season) may be helpful for the reader as the similar observations have been made before.

David Bierbach

Bierbach, D., S. Oster, J. Jourdan, L. Arias-Rodriguez, J. Krause, A. M. Wilson and M. Plath (2014). "Social network analysis resolves temporal dynamics of male dominance relationships." *Behavioral Ecology and Sociobiology*: 1-11.

Doran, C., D. Bierbach and K. L. Laskowski (2019). "Familiarity increases aggressiveness among clonal fish." *Animal Behaviour* 148: 153-159.

We are grateful to Dr Bierbach for his positive and constructive feedback. We agree that we should have included these points in our original discussion. We now discuss the potential role of familiarity, social dominance and sex-specific interaction patterns (Lines 301-303, 314-319). Many thanks for highlighted your work in these areas.

Reviewer 2

Review notes for “Emergence through time of variation between groups in fish shoal collective motion”

In this study, the authors examine how standard, important, reasonable quantitative measures of collective motion vary in time for shoals of three-spined sticklebacks. The study spans a large set of repeated observations of twelve groups of initially newly subgrouped shoals of eight fish, with observations of 12.5 minutes duration in total (drawn from a larger set of observations of approximately 25 minutes that included a regular feeding regime) of each group made up to twelve times over a period of four weeks. The nature of the data set means that variation in collective motion measures could be examined over relatively short time scales (from subsets of the 12.5 minutes of data recorded per experimental trial) and longer durations (from the repeated observations of groups over a four week period). The authors determined that over both the short and long time scales, cohesion (quantified by the area of the convex hull that contained all group members), group order (quantified by polarisation, a measure of overall group alignment),

the speed of the group centroid and a proxy for information transfer (the maximum cross-correlation in speed) all diminished as time from the start of observations increased. I think these are extremely interesting results! In addition, the authors identified variation across groups in both measures of collective motion, and how these measures varied over time, which may have (or likely has) a connection to how the behaviours of individual group members influence and determine overall collective movements.

I really enjoyed working through this paper. I think the paper is well written, the analysis is well done, the results are interesting, and the results point to multiple avenues for further potential research. As such, I happily recommend that the manuscript be accepted for publication subject to addressing the following minor considerations.

We are pleased that the reviewer enjoyed reading our manuscript and we thank them for their positive and helpful comments. Please see our point-by-point responses below.

Main text

Lines 1-2. I'm not sure that the title clearly captures the content of the paper as well as it could. Maybe something like "Group cohesion, order and information transfer diminish overtime in fish shoals", or more specifically "Group cohesion, order and information transfer diminish over time in shoals of three-spined sticklebacks (*Gasterosteus aculeatus*)" could be considered as alternatives?

Based on the reviewer's suggestions, we have changed the title to 'Collective motion diminishes, but variation between groups emerges, through time in fish shoals'. We agree that the overall reduction in cohesion and group order is an important result to highlight. Nonetheless, the most novel aspect of our study is the result that variation between groups is magnified over time so we have retained this finding in the title as well.

Line 112. It's noted that fish were not sexed. Has there been any work to examine potential differences in the movement behaviour of male and female sticklebacks? Might the observed differences across groups be related to male/female composition?

This is an interesting question. In foraging experiments, male three-spined sticklebacks show greater exploration tendencies compared to females spending more time out of cover (e.g. King et al. 2013), however it remains unclear how this would play out in a free-swimming group context where fish can exchange social information. To our knowledge, no studies have directly examined differences in the collective motion of free swimming shoals of three-spined sticklebacks that vary in their sex ratios. Herbert-Read et al. (2017) examined the movement behaviour of male and female same-sex shoals of Trinidadian guppies and found evidence for differences in some traits (e.g. shoal width and length) but not in others (e.g. individual speeds, centroid speeds, the proportion of time shoals spent in a highly polarised state). Reviewer 1 commented on a potential role for sex related interaction patterns although our method of individual assignment to groups was selected to minimise between-group heterogeneity. We have added an additional comment to the discussion to acknowledge this as a potential factor (Lines 301-303).

Line 113. In relation to the group sizes, has any work been done on three-spined sticklebacks that examines the retention of individual movement traits or tendency to conform as a group size varies? I know that idiosyncrasies of the movement patterns of individuals in another species, eastern mosquitofish, become harder to identify as group size increases (see for example, Herbert-Read et al. (2013), Proc R Soc B 280(1752): 20122564) – could the

variation across groups be explained by the groups being of a size that group members are not(yet) conforming to each other? Perhaps heterogeneity of group members in general movement behaviour (including the underlying dynamic rules that govern collective motion) could explain the variation across groups that has been observed? Theoretical support for this idea comes from Romey (1996), *Ecological Modelling* 92:65-77, and empirical support from Jolles et al. (2017), *Current Biology* 27:2862-2868.

We are not clear how an absence of conformity could explain why the groups showed different rates of change in behaviour over time. More likely, conformity could explain the emergence of between-group variation over time either because individuals converge towards the mean behaviours of the group (e.g. Herbert-Read et al. 2013) or towards the most extreme phenotypes (e.g. Jolle et al. 2017). We thank the reviewer for highlighting that we did not previously explicitly discuss social conformity and have now added this (Lines 296-298).

Lines 175-181 and bottom of Table 1 (on page 13 of the review version of the manuscript). Given that the results relating to switches in leadership taking into account time interval and day were not reliable, might it be reasonable to examine durations spent in the leadership position (the front of the group, which is a completely reasonable hypothesis) via survival analysis (for example, Kaplan-Meier survival curves with appropriate confidence intervals for visualisation, and log-rank tests for significance)?

We thank the reviewer for the suggestion. We did not consider using survival analysis for the switches in leadership; it is an interesting idea and would be useful in some data sets, but because durations spent in the leadership position are rarely censored in our study, survival analysis is not appropriate. The durations spent in the leadership position are only censored if the group is polarised at the end of the 2.5 minute interval; if another individual occupies the front position, or if the group goes into a non-polarised state, the leadership duration of an individual ends and is not censored data. There is a mean of 7.2 switches per interval, and the groups are polarised 56% of the time, thus the proportion of censored leadership durations is low.

Lines 314-315. I think there is a missing right parenthesis.

We thank the reviewer for spotting this and have corrected the error.

Supplementary materials and methods

I think the presentation of some of the equations could be improved.

At the second line under *Quantification of traits*, I suggest writing the individual coordinates as $\square x_i(t), y_i(t) \square$ with t measured in time, rather than time steps.

(i) I suggest writing the equation for the polarisation as:

so that the fact that the direction vectors are a function of time is unambiguous.

We agree with the reviewer that incorporating time t improves the presentation of the equations and have updated the ESM accordingly.

(ii) Given that the authors have established notation for the duration between time steps/tracked video frames as Δt , I suggest writing the equation for the centroid speed as:

I suggest rewriting the equation for the group centroid slightly as:

We have corrected this as suggested.

(vi) I'm not sure about the formulation used by the authors here, although it's possible that it could just be due to some typos. The method for determining the front-to-back position of group members first involves for each individual, indexed i , determining the component of the vector from the group centroid to the position of individual i in the direction of the group velocity vector (as summarised by the equations for $\cos \theta$ and d). Based on the equation for the group speed given at (ii) in the supplementary methods, the authors (implicitly) used a backwards difference approximation for the x - and y -components of the group velocity vector (which is valid). Where I think there is an error is that the position references for both the group centroid and the position of individual i should be taken from the same time, t , but the equation for the dot product ($\cos \theta$) suggests that the authors constructed the vector from the group centre at step $t - 1$ (or time $t - \Delta t$) to the position of individual i at the next time step or just time) t . If this is what the authors did, then I suggest that these calculations might have to be redone/checked using the equations I suggest next. If this is just a typo, then I think that the equation for $\cos \theta$ could be rewritten as

if my suggestion above about writing t as time rather than time step is applied. The rewriting of the coordinates of the fish is not needed again, if defined as I suggest at the beginning of the *Quantification of traits* section, and the equation for d could be written as:

We thank the reviewer for spotting the error in our original formula. We have checked the code to ensure that there were no mistakes. This was indeed a typo and we have updated the definitions of $\cos \theta$ and d in the ESM as suggested.

The equation for the mean nearest neighbour distance on page 3 does not seem correct. Perhaps it could be written as:

Many thanks for the suggested change in notation. We agree that this makes the NND clearer and have updated the formula as suggested.